# Advance care planning information intervention for persons with mild dementia and their family caregivers: Impact on end-of-life care decision conflicts

Hsiu-Li Huang[1]⊙*, Wei-Ru Lu[2]⊙, Chien-Liang Liu[3,4], Hong-Jer Chang[1]

**1** Department of Long-Term Care, College of Health Technology, National Taipei University of Nursing and Health Sciences, Taipei, Taiwan, **2** Department of Nursing, Sijhih Cathay General Hospital, New Taipei City, Taiwan, **3** Dementia Center, Taipei City Hospital, Taipei, Taiwan, **4** University of Taipei, Taipei, Taiwan

⊙ These authors contributed equally to this work.
* hsiuli@ntunhs.edu.tw

**Data Availability Statement:** All relevant data are within the manuscript and its Supporting Information files: S1 File. Minimal anonymized dataset.

## Abstract

Persons with dementia are at high risk for loss of decision-making ability due to increased cognitive decline as the disease progresses. Participation in advance care planning (ACP) discussions in the early stages of dementia is crucial for end-of-life (EoL) decision-making to ensure quality of EoL care. A lack of discussions about ACP and EoL care between persons with dementia and family caregivers (FCGs), can lead to decisional conflicts when persons with dementia are in the later stages of the disease. This study explored the effects of a family-centered ACP information intervention among persons with dementia and FCGs. The study was conducted in outpatient clinics in Taiwan. Participants were dyads (n = 40) consisting of persons diagnosed with mild cognitive impairment or mild dementia and their FCGs. A one-group, pretest–posttest, pre-experimental design was employed. The intervention was provided by an ACP-trained senior registered nurse and was guided by ACP manuals and family-centered strategies. Outcome data were collected with four structured questionnaires regarding knowledge of end-stage dementia treatment, knowledge of ACP, attitude towards ACP, and EoL decisional conflict about acceptance or refusal of cardiopulmonary resuscitation, ventilators, and tracheostomy. Paired t tests compared differences between pre-intervention data and 4-weeks' post-intervention data. The intervention resulted in significant improvements among persons with dementia and FCGs for knowledge of end-stage dementia treatment ($p$ = .008 and p < .001, respectively), knowledge of ACP (both $p$ < .001), and significant reductions in decisional conflicts (both p < .001). Scores for positive and negative attitude toward ACP did not change for persons with dementia; however, there was a reduction in negative attitude for FCGs ($p$ = .001). Clinical care for persons with dementia should incorporate ACP interventions that provide knowledge about EoL dementia care using family-centered care strategies that facilitate regular and continuous communication between FCGs, persons with dementia, and medical personnel to reduce decisional conflicts for EoL care.

**Funding:** This study received a fund from the Ministry of Science and Technology of Taiwan (MOST 106-2314-B-182 -006 -MY3). The funder had no role in study design, data collection, and analysis, decision to publish, or preparation of the manuscript.

**Competing interests:** The authors have declared that no competing interests exist.

## Introduction

The prevalence of dementia is increasing due to the rapid growth of the older adult population. At present, approximately 50 million people have dementia worldwide, with an average increase of one new case of dementia every 3 seconds, which doubles every 20 years. The estimated population of people affected by dementia will reach 131.5 million people in 2050 [1]. In Taiwan, the prevalence of dementia among people over 65 years is 8.04% [2], affecting more than 290,000 people [3]. The life expectancy of an individual following a diagnosis of mild dementia ranges from 2 to 7 years; for individuals with moderate to severe dementia, life expectancy ranges from 1.5 to 2.5 years [4, 5]. The World Health Organization reported that between 2000 and 2016 deaths due to dementia doubled, making dementia the fifth leading cause of death globally, compared with a ranking of fourteenth in 2000 [6].

As the disease advances, the trajectory for persons with dementia is characterized by a gradual decline in memory, cognition, activities of daily living (ADLs), and decision-making abilities. With the exception of dementia due to an underlying illness, such as chronic hypertensive encephalopathy, hepatic dysfunction, or vitamin deficiency, which account for less than 10% of all dementias, the decline is irreversible and cognitive abilities decrease with the progression of the disease [7, 8]. Currently, there is no cure for dementia, thus it is considered to be an end-stage disease requiring hospice-palliative care in the late stages [9]. Persons with dementia with late-stage or end-stage disease have profound functional deficits, which make them dependent on fulltime care and supervision by others [10, 11]. These deficits during end-stage dementia can require EoL life-sustaining medical treatments such as cardiopulmonary resuscitation (CPR), placement on a ventilator, or a nasogastric feeding tube [12]. If persons with dementia have not established EoL care preferences prior to developing late-stage dementia, family members are often challenged with the difficulty of acting as an agent and making EoL decisions for persons with dementia [13, 14]. Moreover, systematic literature reviews revealed that proxies were often unable to accurately predict patients' treatment preferences, especially for people with stroke or dementia [15, 16].

Advance care planning (ACP) allows individuals to describe their preferences regarding EoL care. For persons with dementia it is critical that ACP be established with health professionals and family members before they lose their ability to make independent decisions. ACP is regarded as optimal for realization of patient autonomy with regards to palliative care. ACP can enhance discussions of EoL care between patients and healthcare professionals, as well as family caregivers, which is conducive to improving the consistency between patient expectations and care provisions [17]. A randomised controlled study found when older hospitalized adults participated in ACP discussions with family members about EoL care preferences, family members were significantly more satisfied with EoL treatment decisions, and there was a significant decrease in anxiety and depression in relation to decision making [18].

The increase in the prevalence of dementia has drawn attention to decisions about EoL care for persons with dementia, with a focus on conducting discussions about ACP as early as possible, before there is a significant decline in cognitive functions [9, 19]. Taiwan officially implemented the Patient Autonomy Act in early 2019 and listed ACP as a necessary legal procedure before signing advance directives [20]. However, EoL discussions often exclude persons with dementia and the topic of death continues to be taboo in Taiwanese culture. Family members (particularly children or next-of-kin) are responsible for decisions, and the autonomy of persons with dementia is rarely considered. As a result, persons with dementia are reluctant to express their wishes about EoL care to medical personnel or family members, which reduces their participation in decision-making [21]. Many family caregivers (FCGs) experience difficulty and conflict regarding life-sustaining treatment decisions when they do not understand

EoL care preferences of persons with dementia. Therefore, developing effective strategies to protect patient autonomy in decisions regarding EoL care for persons with dementia and reducing conflicts between patients and family members are urgent challenges in clinical care.

Previous studies indicate that lack of knowledge regarding consequences of the disease trajectory, the benefits/risks of EoL treatments, and a negative attitude toward ACP are key barriers to completing ACP [22, 23]. Thus, elements of ACP should be clarified to help patients with the decision-making process when their cognitive abilities are intact, which should include: clarifying a patient's understanding of their illness and treatment options; understanding their preferences and goals for EoL care; identifying their values and beliefs; and reducing decision-making conflicts [18, 24]. However, the strategies regarding effective delivery of ACP interventions for persons with dementia remain a challenge for healthcare providers.

ACP directives emphasize the value of an individual's personal autonomy and preferences. Thus, in many western countries, family members in many western countries are not responsible for decisions about ACP. However, family harmony and interdependency are highly valued in Taiwanese culture, particularly when facing end-of-life care decisions [25], which makes it less likely an individual will initiate an advance directive. The increase in the prevalence of dementia in the general population has drawn attention to the need for shared decision-making between family caregivers and persons with dementia as a component of dementia care [26, 27]. A systematic review of the literature determined the person most likely to be responsible for initiating ACP for persons with dementia is a family member, and family members are also central to facilitating participation of persons with dementia [28]. Studies have shown educating persons with dementia about ACP in the early stages of the disease through discussions can increase participation [22, 29]; one of these studies invited family members to participate in the discussions because the persons with dementia were nursing home residents [22]. An intervention that not only provides information about EoL treatments and ACP for persons with dementia, but also involves FCGs, can increase the willingness of FCGs to act as an agent for persons with dementia and develop an ACP [23, 24, 30]. However, the effectiveness of this type of intervention for FCGs and persons with dementia conducted when persons with dementia are in the early stages of dementia has not been examined. Therefore, this study developed a family-centered ACP information intervention that included individuals with mild dementia and their FCGs. The effect of the intervention was measured using four structured questionnaires regarding knowledge of end-stage dementia treatment, knowledge of ACP, positive and negative attitudes towards ACP, and decisional conflicts for EoL care. The findings could be a reference for promoting ACP for persons with dementia.

## Materials and methods

### Design and study population

The study was conducted from March 2019 to March 2020. Dyads of persons with mild dementia and their FCGs were recruited from outpatient clinics of regional teaching hospitals, dementia care centers, or community care centers located in Taipei City and New Taipei City, Taiwan. The inclusion criteria for persons with dementia were: > 55 years of age; diagnosed with mild dementia or mild cognitive impairment based on a score of 0.5–1 on the Clinical Dementia Rating (CDR) scale or a score $\geq$ 18 on the Mini-Mental State Examination (MMSE); had no established ACP; and able to communicate in Mandarin or Taiwanese. Inclusion criteria for family caregivers were: > 20 years of age and able to communicate in Mandarin or Taiwanese. Both persons with dementia and their primary caregivers had to agree to participate in the study.

Sample size for this study was estimated for F tests with repeated measurements and analysis of variance, where α = 0.05, power = 0.85, and effect size = 0.25 to 0.30 [31], which was estimated to be at least 33 dyads. However, attrition rate has been reported to be 15% for relevant studies on ACP [23, 30], therefore we estimated a total of 40 dyads were required.

The study was conducted in accordance with the Declaration of Helsinki and approved by the Taipei City Hospital Research Ethics Committee (TCHIRB-10605118-E) and consent was also provided by the outpatient clinics, as they do not have independent review boards. Persons with mild dementia or cognitive impairment were informed of the study by flyers posted in the clinics and referrals from nurses, clinicians, and leaders of dementia support groups. The researchers explained the design and purpose of the study to all interested persons with dementia and their FCGs, and assured them of anonymity of the data. All participants provided written informed consent prior to the pretest; persons with dementia provided consent for collection of medical data from their charts. We also invited caregivers to confirm that the person with dementia understood the purpose of this study and were willing to participate. If any person with dementia showed distress, appeared distracted, or provided an answer that was not relevant to the question a reminder about the question or a short break was offered. If the response of the participant did not improve, the interview ceased. If either the person with dementia or their caregiver withdrew or terminated participating in the research, all data collection ceased for the dyads.

## The ACP information intervention

Prior to initiation of the intervention, pretest data were collected from participants using four structured questionnaires, described below. The ACP information intervention was conducted in two parts by a senior nurse who was ACP-trained. The first part of the intervention provided persons with dementia and caregivers with an ACP manual, titled, *Be My Own Master*: *Talk about Advance Care Planning for Dementia*. The manual included both text as well as pictorial aids, which have been shown to significantly improve understanding and decision-making for persons with mild to moderate dementia [32]. The nurse explained the contents of the manual, which included descriptions about the symptoms of end-stage dementia and the common EoL life-sustaining medical treatments, such as CPR, machine ventilation, tube feeding, intravenous infusion, and antibiotics. The manual also provided details about the benefits and risks of the treatments, as well as the process of establishing ACP and the regulations involved. The second part of the intervention involved family-centered strategies for developing ACP and the nurse facilitated open discussions between the persons with dementia and their FCGs. Dyads were encouraged to communicate their thoughts and expectations about EoL care, and to describe any uncertainties about the process; the nurse intervener provided any needed clarifications. The intervention lasted approximately 60 min. Four weeks after completing the intervention a posttest was conducted using the four structured questionnaires. Data collection was executed by a trained research assistance.

## Sociodemographic and clinical characteristics

Structured questionnaires were used to collect data about demographic and clinical characteristics of the persons with dementia, and demographic characteristics of the FCGs. Demographic data for both groups included age, gender, level of education and marital status. Clinical data for the persons with dementia included the presence of other diseases, months since diagnosis of dementia, awareness of their diagnosis of dementia. Demographic data for family caregivers included religion, relationship to the person with dementia, the number of daily caregiving hours, and their self-reported health status (poor, good, excellent). Level of

cognitive ability was determined with scores on the CDR and MMME, which were collected from participants' charts.

## Barthel index scale

The Barthel index scale evaluated activities of daily living (ADLs) for person with dementia as a measure of functional abilities [33], which includes eating, personal hygiene, toileting, bathing, dressing and undressing, bowel and bladder control functions, and mobility. Total scale scores range from 0–100 points; higher scores indicate a higher level of self-care ability. The scale has been shown to have satisfactory reliability and validity for persons with dementia [2].

## Pretest and posttest measures

**Knowledge of end-stage dementia treatment.** This scale assessed participants' recognition of the symptoms of end-stage dementia and the effectiveness of life-sustaining treatments for EoL care for persons with end-stage dementia. Persons with dementia and FCGs were asked eight questions (single [yes/no] and multiple-choice) about their knowledge of end-stage treatments, which included CPR, nasogastric feeding tubes, intravenous therapy for dehydration, and antibiotics. The content validity index (CVI) for the instrument was 0.96; reliability of the scale was determined with the Kruder-Richardson (KR) formula, which was 0.65 [34]. The research team added nine questions to the original scale about symptoms of severe dementia. Both persons with dementia and FCGs were reminded that if they were not sure if the statement was true or false to please answer "unknown". Items were scored as true (1) and, false (0), or unknown (0). The total score ranged from 0–17; a higher score indicated greater knowledge of dementia and EoL care. In this study, the KR value for persons with dementia and FCGs were 0.83 and 0.81 respectively, indicating an adequate and acceptable homogeneous test [35]. The ranges of item difficulty and item discrimination for persons with dementia (0.27–0.55 and 0.45–1.00, respectively) and FCGs (0.27–0.73 and 0.27–0.91, respectively) were within acceptable ranges [36].

**Knowledge of ACP.** Participants' knowledge of ACP was assessed by the researcher-developed Knowledge of ACP scale. Six questions ask about knowledge of the Patient Autonomy Act [20], advance directives, ACP, advance hospice-palliative and life-sustaining decisions, advance health care agency, and decisions to not perform CPR. Questions were scored on a 4-point Likert scale: 0 = I have never heard of this; 1 = I have heard of this, but am not sure what it is; 2 = I have heard about this and know about the content; and 3 = I knew and signed the content. Total scores ranged from 0–15 points; higher scores indicated more knowledge about ACP. In this study, the Cronbach's alphas for persons with dementia and FCGs were 0.83 and 0.89, respectively.

**Advance Care Planning Attitude (ACPA).** The ACPA scale was used to evaluate beliefs and feelings regarding ACP. The 17-item ACPA scale was modified from a scale measuring positive and negative attitudes towards Taiwan's Hospice-Palliative Care Act [25]. The original scale included a total of 14 items describing the pros (positive attitudes, 8 items) and cons (negative attitudes, 6 items) of Taiwan's Hospice-Palliative Care Act. Our research team added one pro-item related to increasing communication about ACP between patient, family and the medical team and two con-items related to ACP procedures and payments. The 17 items were scored on a 5-point Likert scale from (1) strongly disagree to (5) strongly agree. Scores for the positive subscale of the ACPA ranged from 9–45 points; higher scores indicated a more positive attitude. Scores for the negative subscale of the ACPA ranged from 8–40 points; higher scores indicated a more negative attitude. The Cronbach's alpha for the two subscales were

0.90 and 0.85, respectively [25]. In this study, the Cronbach's alpha for the total score was 0.96 and 0.78 for persons with dementia and FCGs, respectively.

**Decisional conflict for EoL care.**   Personal conflicts about decisions for EoL care were measured with a modification of the Decisional Conflict Scale (DCS) developed by O'Connor (1995), which was developed to measure uncertainty about healthcare related decisions, factors that contribute to the uncertainty, and a person's perception of the effectiveness of their decisions [37]. The 16-item DCS includes five subscales: informed (3 items), values clarity (3 items), support (3 items), uncertainty (3 items), and effective decision (4 items). The DCS for EoL care was modified for persons with dementia and FCGs. Each item of the scale was a question about EoL care decision options to either accept or forgo: CPR, machine ventilation, and tube feeding. The question format is easier for participants with limited reading skills [38]. Items were scored on a 5-point Likert scale: from (1) strongly agree to (5) strongly disagree. If the participant indicated that they had no actual decision-making ability, the research assistant asked them to indicate what their choice would be if they needed to make an EoL care decision right now. For example, a question on the subscale of effective decisions, states "I feel I have made an informed choice about EoL care for myself (person with dementia)/or for my relative (FCG)". If the participant was unwilling to make a hypothetical choice, the question was marked as not applicable, and the calculated score was weighted based on the total answers for the subscale. The total scores were transformed to a scale of 0 (no decisional conflict) to100 (extremely high decisional conflict). Internal consistency for related studies was 0.72–0.86 [37, 38]. In this study, the total Cronbach's alphas for persons with dementia and FCGs were 0.96 and 0.98, respectively.

## Statistical analysis

SPSS 22.0 software for Windows was used for statistical analysis. Descriptive statistics included frequency, percentage, mean, and standard deviation (SD). The effect of the intervention was determined with a paired sample t test and effect size by analyzing the difference in pre-intervention and post-intervention effect scale scores for knowledge of end-stage dementia treatment, the knowledge of and attitude toward ACP, and decisional conflicts for EoL care. A level of $p < .05$ was considered significant.

## Results

A total of 104 persons with dementia and FCGs were eligible to participate in the intervention. However, 26 dyads were not included in the final analysis because the FCG did not agree to participate (n = 16), contact information was incorrect (n = 6), or the intervention was not completed due to hospitalization of the patient (n = 1) or schedule conflicts (n = 3). Therefore, a total of 78 persons (40 dyads) completed the intervention; two caregivers cared for two persons with dementia, thus they were paired with the two persons with dementia repeatedly for a total of 40 persons with dementia and 38 FCGs.

Characteristics of the persons with dementia and FCGs are shown in Table 1. Most PWDs were married (73%); more than half were female; most had been informed of or knew of their diagnosis of dementia (68%). The mean score on the MMSE was 20.5 (SD = 4.83); 35% scored 0.5 on the CDR and 65% scored 1. The mean age of the FCGs was 56.9 years (SD = 12.32); 84% were female; 76% were married, and 82% had a high school education or higher. Most FCGs were the child of the person with dementia (63%); 45% identified their religion as Buddhist or Taoist. Self-reported health status was considered to be good by 60% of FCGs and the mean number of weekly caregiving hours was 13.0 (SD = 9.30).

**Table 1. Sociodemographic characteristics of persons with mild dementia and family caregivers.**

| Characteristic | Persons with dementia (n = 40) | | Family caregivers (n = 38) | |
|---|---|---|---|---|
| | n (%)[a] | Mean (SD) | n (%)[a] | Mean (SD) |
| Age | | 77.5(8.24) | | 56.9(12.32) |
| Gender | | | | |
| Male | 19 (48) | | 6 (16) | |
| Female | 21 (53) | | 32 (84) | |
| Marital status | | | | |
| Married | 29 (72) | | 29 (76) | |
| Single | | | 7 (18) | |
| Widowed | 11 (28) | | 2 (5) | |
| Educational level | | | | |
| None | 3 (8) | | | |
| ≤ Junior high school | 16 (40) | | 7 (18) | |
| ≥ High school | 21 (53) | | 31 (82) | |
| Characteristics for persons with mild dementia only | | | | |
| Other illnesses or diseases | | | | |
| None | 14 (35) | | | |
| 1 to 2 | 20 (50) | | | |
| 3 to 4 | 6 (15) | | | |
| Months since diagnosis (range = 1–54) | | 27.0 (23.50) | | |
| Informed or aware of diagnosis | | | | |
| Yes | 27 (68) | | | |
| No | 13 (33) | | | |
| MMSE score (range = 7–29) | | 20.5 (4.83) | | |
| CDR score (mild dementia range, 0.5–1.0) | | | | |
| 0.5 points | 14 (35) | | | |
| 1.0 point | 26 (65) | | | |
| ADL (Range = 0–100) | | 95.1 (10.71) | | |
| Characteristics for family caregivers only | | | | |
| Religion | | | | |
| None | | | 14 (37) | |
| Buddhism or Taoism | | | 17 (45) | |
| Christian | | | 6 (16) | |
| Other | | | 1 (3) | |
| Relationship to patient | | | | |
| Spouse | | | 13 (34) | |
| Child | | | 24 (63) | |
| Daughter-in-law | | | 1 (3) | |
| Self-reported health status | | | | |
| Poor | | | 4 (11) | |
| Good | | | 23 (60) | |
| Excellent | | | 11 (29) | |
| Number of daily caregiving hours | | | | 13.0 (9.30) |

Note: SD = standard deviation; MMSE = Mini-Mental State Examination; CDR = Clinical Dementia Rating scale; ADL = activities of daily living.

[a] Due to rounding, some totals may not correspond with the sum of the separate figures.

### Effects of the ACP information intervention on persons with dementia

The effect of the ACP information intervention on persons with dementia was determined by the difference in pretest and posttest scale scores (Table 2). Scores for knowledge of end-stage dementia treatment and ACP improved significantly (t = −2.79, p = .008, effect size = 0.5; and t = −4.10, p < .001, effect size = 0.6, respectively). No significant change was observed in either the positive (33.00 → 34.98) or negative (23.03 → 22.93) attitude towards ACP. The mean total score for posttest decisional conflicts in EoL care was significantly lower than pretest for persons with dementia (49.88 → 35.11, t = 4.76, p < .001, effect size = −0.8). All five subscale scores were also significantly lower: informed (55.00 → 40.83, t = 2.88, p < .01, effect size = -0.06), values clarity (58.96 → 41.46, t = 3.76, p < .01, effect size = -0.08), support (48.54 → 36.11, t = 3.86, p < .001, effect size = -0.06), uncertainty (45.63 → 30.42, t = 4.37, p < .001, effect size = -0.067), and effective decision (43.44 → 29.22, t = 4.05, p < .001, effect size = -0.06).

### Effects of the ACP information intervention on family caregivers

The effect of the ACP information intervention, as measured by differences in pretest and posttest scale scores for FCGs, is shown in Table 3. Similar to persons with dementia, scales scores for knowledge of end-stage dementia treatment and knowledge of ACP improved significantly (t = −4.77, p < .001, effect size = 0.8; and t = −8.22, p < .001, effect size = 1.2, respectively). In contrast to a lack of change in positive or negative attitudes towards ACP for persons with dementia, FCGs had a significant decrease in negative scores (24.27 → 20.65, t = 3.73, p = .001, effect size = −0.7). The total score for decisional conflicts decreased significantly after the intervention (37.34 → 27.70, t = 3.89, p < .001, effect size = −0.6). In addition, significant decreases were seen for all five subscale scores, with the largest decreases for informed (34.79 → 23.13, t = 4.06, p < .001, effect size = −0.8) and values clarity (39.58 → 24.79, t = 4.54, p < .001, effect size = −0.9). The smallest decrease was in support (37.29 → 31.46, t = 2.13, p < .05, effect size = −0.3). The decreases for uncertainty and effective decision were 37.50 → 29.58 and 37.50 → 29.06, respectively.

**Table 2. Mean scale scores for Knowledge of End-Stage Dementia (KESD), Advance Care Planning Attitude (ACPA), and decisional conflicts in End-of-Life (EoL) care among persons with dementia, pre- and post-intervention (n = 40).**

| Scales and subscales | Pre-intervention Mean ± SD | Post-intervention Mean ± SD | t | P | 95% CI | Effect size |
|---|---|---|---|---|---|---|
| Knowledge of ESD treatment (range = 0–17) | 6.38 ± 4.16 | 8.75 ±4.74 | -2.79 | .008 | -4.07 ~ -0.65 | 0.5 |
| Knowledge of ACP (range = 0–15) | 2.95 ± 3.49 | 5.33 ± 4.20 | -4.10 | < .001 | -3.54 ~ -1.20 | 0.6 |
| ACPA | | | | | | |
| Positive (range = 9–45) | 33.00 ± 6.09 | 34.98 ± 7.01 | -1.56 | .126 | -4.52 ~ 0.57 | 0.3 |
| Negative (range = 8–40) | 23.03 ±4.02 | 22.93 ± 3.06 | 0.12 | .908 | -1.63 ~ 1.83 | -0.02 |
| Decisional conflicts in EoL Care | | | | | | |
| Total score (range = 0–100) | 49.88 ± 21.03 | 35.11 ± 16.62 | 4.76 | < .001 | 8.49 ~ 21.05 | -0.8 |
| Subscales | | | | | | |
| Informed (range = 0–100) | 55.00 ± 22.55 | 40.83 ± 25.09 | 2.88 | .006 | 4.23 ~ 24.11 | -0.6 |
| Values clarity (range = 0–100) | 58.96 ± 23.30 | 41.46 ± 23.07 | 3.76 | .001 | 8.10 ~ 26.90 | -0.8 |
| Support (range = 0–100) | 48.54 ± 19.33 | 36.11 ± 19.70 | 3.86 | < .001 | 5.93 ~ 18.94 | -0.6 |
| Uncertainty (range = 0–100) | 45.63 ± 23.79 | 30.42 ± 19.93 | 4.37 | < .001 | 8.17 ~ 22.24 | -0.7 |
| Effective decision (range = 0–100) | 43.44 ± 25.43 | 29.22 ± 19.01 | 4.05 | < .001 | 7.12 ~ 21.32 | -0.6 |

Note: SD = standard deviation; CI = confidence interval (lower, upper); Effect size = Cohen's d.

**Table 3. Mean scale scores for knowledge of End-Stage Dementia (ESD) treatment, Advance Care Planning Attitude (ACPA), and decisional conflicts in End-of-Life care (EoL) care among family caregivers pre- and post-intervention (n = 40).**

| Scales and subscales | Pre-intervention Mean ± SD | Post-intervention Mean ± SD | t | P | 95% CI | Effect size |
|---|---|---|---|---|---|---|
| Knowledge of ESD treatment (range = 0–17) | 10.14 ± 3.62 | 13.13 ± 3.38 | -4.77 | < .001 | -4.24 ~ -1.71 | 0.8 |
| Knowledge of ACP (range = 0–15) | 6.70 ± 3.68 | 10.65 ± 2.50 | -8.22 | < .001 | -4.92 ~ -2.97 | 1.2 |
| ACPA | | | | | | |
| Positive (range = 9–45) | 40.08 ± 4.47 | 39.08 ± 4.59 | 1.47 | .149 | -0.37 ~ 2.35 | -0.2 |
| Negative (range = 8–40) | 24.27 ± 4.12 | 20.65 ± 5.26 | 3.73 | .001 | 1.65 ~ 5.58 | -0.7 |
| Decisional conflicts in EoL care | | | | | | |
| Total score (range = 0–100) | 37.34 ± 16.43 | 27.70 ± 13.34 | 3.89 | < .001 | 4.63 ~ 14.67 | -0.6 |
| Subscale scores | | | | | | |
| Informed (range = 0–100) | 34.79 ± 19.14 | 23.13 ± 11.08 | 4.06 | < .001 | 5.85 ~ 17.48 | -0.8 |
| Values clarity (range = 0–100) | 39.58 ± 20.22 | 24.79 ± 13.80 | 4.54 | < .001 | 8.19 ~ 21.39 | -0.9 |
| Support (range = 0–100) | 37.29 ± 16.77 | 31.46 ± 16.61 | 2.13 | .039 | 0.30 ~ 11.37 | -0.3 |
| Uncertainty (range = 0–100) | 37.50 ± 17.60 | 29.58 ± 16.77 | 2.64 | .012 | 1.85 ~ 13.98 | -0.5 |
| Effective decision (range = 0–100) | 37.50 ± 18.07 | 29.06 ± 19.43 | 2.82 | .007 | 2.39 ~ 14.49 | -0.4 |

Note: SD = standard deviation; CI = confidence interval (lower, upper); Effect size = Cohen's d.

## Discussion

To our knowledge, this study is the first family-centered ACP intervention model in Asia. Although ACP emphasizes the patient's autonomy and is designed to be person-centered, family members are usually expected to be proxies in Chinese countries. If the FCGs do not understand the content of an ACP for the patient, they are often unable to adhere the patient's preferences. The Taiwanese Patient Right to Autonomy Act, Article 9 stipulates that "the declarant, at least one relative of first or second degree of affinity, and the health care agent shall participate in the advance care planning" [39]. The purpose of this procedure is to increase communication and promote a harmonious relationship between the patient, family and medical professionals. Therefore, the family-centered ACP intervention model may be more suitable for countries with a Chinese culture that is family-centered, such as Taiwan.

The ACP information intervention demonstrated a significant improvement in knowledge of end-stage dementia treatments, knowledge of ACP, and decisional conflicts regarding EoL care for persons with mild dementia and their FCGs. The research suggests that inclusion of both persons with dementia and primary FCGs is essential in this family-centered ACP model. The use of ACP manuals and pictures helped guide persons with dementia and their FCGs in understanding symptoms of end-stage dementia, common life-sustaining medical treatments, and ACP. Persons with dementia were able to express their thoughts and feelings regarding their own EoL-care involving life-sustaining treatment, which lead to discussions and new knowledge acquisition between the dyads. The following sections discuss the effectiveness of the intervention on knowledge, attitude, and decisional conflicts.

### Effects of the Intervention on knowledge of end-stage dementia treatment and advance care planning

Providing information about end-stage treatment for dementia and knowledge about the components of ACP are crucial to helping persons with dementia make informed decisions about ACP [40]. After the intervention, knowledge of end-stage treatment and ACP

significantly improved for FCGs, which is consistent with findings of other studies [23, 24]. The mean scores for persons with dementia also improved significantly for these variables following the intervention. However, the mean scores for knowledge of end-stage dementia treatments and ACP were higher for FCGs than persons with dementia both before and after the intervention. One explanation for these higher scores may be because the education level of ≥ high school was 83% for FCGs compared with only 50% for persons with dementia. In addition, although the persons with dementia had only mild dementia, their mean age was 78 years, thus their older age along with their dementia might have impacted their memory and concentration, making it more difficult to absorb the information offered during the intervention. However, including visual aids, such as pictures, during the intervention process can deepen the impressions and reactions of persons with dementia significantly for decisions about EoL care [32], which might explain the significant improvement for the persons with dementia, even though their scores were lower than those for the FCGs. In addition, the persons with dementia in our study had the opportunity to communicate with FCGs and the nurse facilitator, which has been shown to provide persons with mild dementia the ability to continue to participate in learning and discussions regarding their personal affairs [41].

## Effects of the intervention on attitude toward advance care planning

Post-intervention scores for either positive or negative attitudes toward ACP were not significantly different for the persons with dementia. For FCGs, there was also no significant difference in positive attitudes towards ACP, however, scores for negative attitude significantly decreased. The decline in negative attitude among FCGs following the intervention suggests most caregivers maintained an open attitude during the discussions. In contrast, the persons with dementia may have been less able to discuss their concerns due to age- and disease-related factors. Other studies have shown persons with dementia are less interested in abstract problems unrelated to urgent needs [42–44].

There are several explanations for the lack of significant differences in positive attitudes following the intervention for both the persons with dementia and FCGs. One explanation is that the participants may have had a positive attitude before the intervention, and the ceiling effect yielded no significant difference after the intervention. A second explanation might be due to the ACP attitudinal scale used in this study, which was developed previously to reflect attitudes towards the Hospice-Palliative Care Act [25] legislated in Taiwan in 2000. The Patient Right to Autonomy Act was not officially launched until January 2019, therefore participants may not have been familiar enough with the new regulations involving ACP to express an opinion. If the participants did not express any positive or negative opinions regarding ACP during the intervention, there was no opportunity for the facilitator to address these attitudes and values and provide any clarification. A third explanation is that attitude represents deeper ideas and values related to life experiences. ACP is a dynamic process, which often requires regular and continuous communication for ACP-specific ideas [45–47].

One explanation for the lack of significant differences in positive attitudes pre- and posttest scores applies specifically to the persons with dementia. This study was a single intervention with a 4-week interval between pretest and posttest. The characteristics of dementia might have made it difficult for the persons with dementia to retain the information provided during the intervention. Increasing the frequency of intervention sessions in the future could allow persons with dementia to gradually adapt to the communication mode of the ACP intervention, which could assist them in retaining the information about ACP.

## Effects of the Intervention on decisional conflicts regarding EoL care

The total score for decisional conflicts in EoL care was significantly reduced following the ACP intervention for persons with dementia and FCGs. This result was consistent with studies showing an ACP intervention significantly reduced EoL care decisional conflicts for FCGs acting as agents for persons with dementia [24, 30]. In our study, the greatest improvement was in the sub-scale score for values clarity, followed by uncertainty. The intervention process facilitated shared decision-making. In the first part of the intervention the persons with dementia and FCGs were taught the symptoms end-stage dementia, as well as the complications, and the benefits and risks of medical treatment at EoL. More importantly, the second part of the intervention encouraged communication between the persons with dementia and FCGs, which provided an opportunity for persons with dementia to clarify what values were important and what their expectations were, and FCGs could share their concerns. Being better informed allowed dyads to weigh the advantages and disadvantages of life-sustaining treatment and their own values, which might have contributed to the reduction in the uncertainty of decision-making.

Although persons with dementia, FCGs, professional caregivers, and researchers agree that ACP discussions be conducted as early as possible [9,18, 48], a review of 67 studies on recommendations from healthcare providers on ACP showed persons with dementia have been excluded from discussions about ACP or from studies on ACP due to assumptions about cognitive problems [49]. Studies on the ACP status of persons with severe dementia in nursing homes often collect data only from family members [24, 30]. This study is unique in that is the first assessment of the effectiveness of an intervention to increase knowledge and facilitate information sharing about ACP between persons with mild dementia and FCGs. We recruited persons with mild cognitive impairment or mild dementia who retained decision-making abilities in order to optimize the initial evaluation of the intervention. During the ACP intervention process, FCGs could help persons with dementia understand that ACP allows one to share their values and wishes prior to losing their decision-making abilities. In addition, because the FCGs were familiar with the characteristics and habits of the persons with dementia, the intervention also helped FCGs identify the key concerns. The companionship and support provided by the FCGs may have helped the persons with dementia feel more at ease, which is a crucial factor for enabling independent decision-making [50–52]. Future studies will evaluate whether this intervention is effective for persons with greater levels of cognitive impairment based on scores on the CDR or MMSE, which could provide information on a cutoff point where the intervention is not helpful.

The process of the intervention was designed to help persons with dementia actively and clearly communicate and interact with FCGs regarding what they see and hear, and convey their values. If FCGs exhibited a relatively passive attitude towards helping the persons with dementia articulate their wishes or concerns, the intervener could use simple questions and answers to provide needed clarifications and guidance. This not only assisted in the organization of thoughts and identification of ideas expressed by the persons with dementia, but also helped the FCGs understand the views of the persons with dementia. This process involved the facilitator repeating questions or content of the discussions to confirm whether the persons with dementia understood what had been discussed, which is something the FCGs might not have the patience to do. Studies have shown the culture of Chinese people does not encourage discussions about EoL care [21, 25]. Therefore, ACP interventions for persons with dementia and FCGs with Chinese cultural backgrounds might have a greater need for nurse facilitators to have strong communication and interpersonal skills.

FCGs had the smallest decrease in the decisional conflict subscale score for support, suggesting the intervention had the least impact on this aspect of EoL conflict for FCGs. This

could have been because other key family members did not contribute to decision making, and therefore there was no support from other family members, such as sharing care costs. Moreover, caregivers continued to experience stress regarding the EoL care decisions, which highlights the essential role Chinese families play when making crucial medical decisions [53–55]. Therefore, continuous communication within families should be encouraged to improve the process of ACP and should include the participation of the family member with dementia, the primary caregiver, and other significant family members.

## Research limitations

Our study had some limitations; therefore, our findings should be viewed with caution. The design of this study was a pre-experimental single-group paired with pretest-posttest design. There was no comparison between an experimental and control group, therefore the evidence is not as strong as if it would be if it were conducted with an experimental or quasi-experimental research design. Due to factors such as limited manpower and time, the collected sample data came only from regional hospitals in northern Taiwan. Next, we only recruited persons with mild cognitive impairment and mild dementia for this study. Excluding persons with moderate dementia prevents the generalization of our findings to all persons with dementia in Taiwan. In addition, our hypothetical situations used to test decision conflicts are not necessarily representative of real-life situations and therefore, the findings should be interpreted with caution. Persons with mild cognitive impairment do not necessarily develop dementia, and the perspectives of persons with mild cognitive impairment regarding decisions about ACP and EoL care might differ from persons with dementia. A more heterogenous sample of persons with dementia should be considered for future studies. Finally, this study was based on dyads of persons with mild dementia and FCGs, which increased the difficulty of obtaining a larger sample size; 14% of FCGs refused to participate in the study (n = 16). Discussions about EoL and death remain culturally sensitive topics in Taiwan and reluctance to discuss EoL might have been the reason these FCGs refused to participate. Therefore, this study may have overestimated the acceptance level for EoL issues.

## Conclusions

This study is the first family-centered ACP intervention model in Asia. Our findings demonstrate that a family-centered ACP information intervention can generate significant improvements in knowledge of end-stage treatment for dementia, knowledge of and attitude towards ACP, and decisional conflicts in EoL care among persons with mild dementia and their family caregivers. In clinical practice, health professionals should include discussions and decisions about the process of ACP as part of family-centered care. We suggest ACP interventions should be structured by training clinicians or nursing professionals to act as facilitators, incorporating ACP into the routine clinic pathway through regular communications with persons with dementia and their FCGS, which could reduce decisional conflicts in EoL care.

## Supporting information

**S1 Dataset. Minimal anonymized dataset.**
(XLS)

**S2 Dataset.**
(XLS)

## Acknowledgments

We would like to thank all the persons with dementia and FCGS who took part in the study. In addition, we thank all the healthcare professionals who referred participants for this study.

## Author Contributions

**Conceptualization:** Hsiu-Li Huang, Wei-Ru Lu, Chien-Liang Liu, Hong-Jer Chang.

**Data curation:** Hsiu-Li Huang, Hong-Jer Chang.

**Formal analysis:** Hsiu-Li Huang, Wei-Ru Lu.

**Funding acquisition:** Hsiu-Li Huang.

**Investigation:** Wei-Ru Lu.

**Methodology:** Hsiu-Li Huang, Wei-Ru Lu, Chien-Liang Liu, Hong-Jer Chang.

**Project administration:** Hsiu-Li Huang.

**Resources:** Chien-Liang Liu.

**Supervision:** Hsiu-Li Huang, Chien-Liang Liu.

**Validation:** Hsiu-Li Huang, Chien-Liang Liu, Hong-Jer Chang.

**Visualization:** Hsiu-Li Huang.

**Writing – original draft:** Hsiu-Li Huang, Wei-Ru Lu.

**Writing – review & editing:** Hsiu-Li Huang.

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
