## [Decision Letter · Decision Letter 0]

3 Aug 2020

PONE-D-20-21167

Advance Care Planning Information Intervention for Persons with Mild Dementia and Their Family Caregivers: Impact on End-of-Life Care Decision Conflicts

PLOS ONE

Dear Dr. Huang,

Thank you for submitting your manuscript to PLOS ONE. After careful consideration by 2 Reviewers and an Academic Editor, all of the critiques of both Reviewers must be addressed in detail in a revision to determine publication status. If you are prepared to undertake the work required, I would be pleased to reconsider my decision, but revision of the original submission without directly addressing the critiques of the two Reviewers does not guarantee acceptance for publication in PLOS ONE. If the authors do not feel that the queries can be addressed, please consider submitting to another publication medium. A revised submission will be sent out for re-review. The authors are urged to have the manuscript given a hard copyedit for syntax and grammar.

**Comments to the Author**

1. Is the manuscript technically sound, and do the data support the conclusions?

Reviewer #1: Yes

Reviewer #2: Partly

2. Has the statistical analysis been performed appropriately and rigorously? 

Reviewer #1: Yes

Reviewer #2: Yes

3. Have the authors made all data underlying the findings in their manuscript fully available?

Reviewer #1: Yes

Reviewer #2: Yes

4. Is the manuscript presented in an intelligible fashion and written in standard English?

Reviewer #1: Yes

Reviewer #2: Yes

5. Review Comments to the Author

Reviewer #1: This is a well timed article addressing serious and pressing concerns in relation to ACP and people with dementia for end of life care. An issue which is increasingly concerning given the increasing prevalence of dementia in Asia and globally. The intervention described serves a purpose even if this raises awareness among FCG's. The obvious detriment is the ability of a person with dementia being able to retain information due to memory loss. The article discusses that studies have shown that interventions regarding ACP can increase awareness and decision making between PWD, FCG's and clinicians.

However, the article omits to mention where there is also a lack of congruency in such discussions (see Jones et alones, K., Birchley, G., Huxtable, R., Clare, L., Walter, T., & Dixon, J. (2016). End of life care: A scoping review of experiences of Advance Care Planning for people with dementia. Dementia: the international journal of

social research and practice. DOI: 10.1177/1471301216676121.

I was also curious about the issue of one off vs ongoing consent that might be expected in a study involving PWD. For example, securing consent and the right to withdraw right up to the stage when data is amalgamated and anonymised and it is no longer possible to withdraw data.

There is also the issue that PWD with dementia having received a diagnosis of early stage dementia as identified by the use of the MMSE and CDR and other information, may in fact find it hard to imagine their future selves as and having to make decisions around oral hydration for example.

On page 25. there is a typo under the heading After effects of th eintervention on decisional conflicts (line 2) 'The result was is'. If the authors could clarify.

I am in agreement with the recommendations that as part of the ACP interventions clinicians and nursing professionals should be trained to facilitate this tool

Overall, I would attend to these queries which are few as this is a good article.

Reviewer #2: There is not much evidence about advance care planning effects in case of early dementia, and therefore a non-controlled pre-post test design does add substantially to the current body of evidence. Most parts of the study are well-designed, individual patient summed scores are being provided, and the article is generally well written.

A main concern is inclusion of persons with no dementia diagnosis but just mild cognitive impairment. This is not only a methodological problem (which may be overcome by sensitivity analyses excluding the persons with MCI). Because the intervention explains disease prognosis to persons who may not have and may not develop the disease, this is also an ethical concern. There was an IRB approval of the local hospital for this intervention study. In some other countries though, monitoring of such study might be as stringent as for drug trials because of the vulnerability of the population and the sensitive nature of advance care planning discussions. Further, did the informed consent form include permission to provide the participants’ data open access?

Below you will find suggestions to improve the manuscript.

Please avoid abbreviating persons by their disease. See for example Alzheimer Australia for language guidelines.

The attitude outcome is very interesting. Unfortunately, it is not clear from the cited reference what the items look like. Ref 29 shows 8 negative and 5 positive items, different from the numbers in the paper.

The study uses the validated decisional conflict scale as a relevant outcome. The knowledge outcomes are not much interesting, as the items of the scales resemble the information provided in the intervention about invasive medical procedures, knowledge that most people lack (e.g. explain CPR and test knowledge about CPR). What is interesting and new though, is that the knowledge test was also completed by persons with dementia. Did you reassure persons with dementia if they felt they failed the test, did you have any protocol for this?

Introduction: “for individuals with moderate to severe dementia, life expectancy ranges from 1.5 to 2.5 years [4-6].” Reference 6, however, depicts a median survival of about 5 years for MMSE 0-18.

Introduction: “the decline is irreversible and becomes more severe” is not accurate. Actually, there is evidence of the reverse, the decline, which is of course a rate becoming less steep towards the end as people can linger on with advanced dementia for a long time (e.g. Gill et al., NEJM) although caution as findings may be related to sensitivity or ceiling effects of some dementia severity measures in the more severe spectrum.

Introduction: please correct “a randomized control study”

Introduction: family members cannot be held responsible for advance care planning, in most countries physicians decisions are ultimately decisive even after shared decision making as they cannot be requested to provide medically futile care

Methods: need to provide a cut off and a proper methodological reference for the Kuder-Richardson statistic

Methods: how did respondents complete the subscales Uncertainty and Effective decision if no decision was taken or if there were more decisions? The items of the other subscales are more general, but the items of these 2 subscales refer to a particular decision. Did you provide any instruction to the respondents, or else, how did you manage while cleaning data?

Results: a total of 86 persons in 40 dyads of whom there were 2 family caring for 2 patients, would result in a total of 78 persons. Please explain how you arrived at 86, does it include persons who refused or did not provide meaningful data? How many did?

Table 1: please avoid too many decimal places that indicate fractions of persons with these modest numbers as it suggests a precision that is not meaningful. For example, 19 of 40 persons is shown as 47.5%. If it were 18 persons, it was 45%. Rounding off to 48% is best as 47.4% or 47.6% is impossible, even 46% is. I know you can still see it in the literature, and unfortunately, the PWD is widespread too, but let’s try to do better and professionalize our area of research.

Results: “positive score increased slightly (33.00 → 34.98)” Please do not overinterpret. The 2 decimals with no SD mentioned in the text are misleading for the magnitude and uncertainty around this “slight effect.”

Results: “was significantly lower for PWDs (49.88)” A more accurate description of the result would be to add the comparator with posttest lower than pretest.

Results: there was a reduction in the uncertainty of decision making. Which decisions? Please add type and number of decisions made to the Results section.

Methods/discussion: What does family-centered mean? Do you mean to say it is not patient-centered even though you choose a population of patients with a mean of about 20 so probably very capable of their preferences being at the center. If this is a cultural issue, please explain as it is relevant for western cultures if the intervention translates to a more individualized context.

Discussion: “A second explanation might be due to the ACP attitudinal scale used in this study, which was a revision of the attitudinal scale of the Hospice-Palliative Care Act [29].” What is the explanation? Please explain.

Discussion: “Our study is unique in that persons with mild dementia who retained decision-making abilities were purposively sampled to participate.” Two observations: not all had dementia. Further, purposive sampling is an expression used mostly for qualitative research to indicate purposeful sampling of a non-representative sample to elicit rich information. I cannot see how this term applies to this particular study other than that there was an inclusion criterion for this dementia-specific dyadic intervention with limitations in and of itself because persons with moderate dementia were excluded and persons with MCI were included.

6. PLOS authors have the option to publish the peer review history of their article (what does this mean?). If published, this will include your full peer review and any attached files.

**Do you want your identity to be public for this peer review?** For information about this choice, including consent withdrawal, please see our Privacy Policy.

Reviewer #1: No

Reviewer #2: **Yes: **JT van der Steen

We look forward to receiving your revised manuscript.

Kind regards,

Stephen D. Ginsberg, Ph.D.

Section Editor

PLOS ONE

2. Please describe in your methods section how capacity to provide consent was determined for the participants in this study. Please also state whether your ethics committee or IRB approved this consent procedure. If you did not assess capacity to consent please briefly outline why this was not necessary in this case.

---

## [Author Response · Author response to Decision Letter 0]

24 Sep 2020

September 14, 2020

Stephen D. Ginsberg PhD

Section Editor, PLOS ONE

Manuscript ID PONE-D-20-21167

Manuscript Title: Advance Care Planning Information Intervention for Persons with Mild Dementia and Their Family Caregivers: Impact on End-of-Life Care Decision Conflicts 

Dear Professor Ginsberg,

The authors appreciate the opportunity to revise our manuscript entitled “Advance Care Planning Information Intervention for Persons with Mild Dementia and Their Family Caregivers: Impact on End-of-Life Care Decision Conflicts”. We have made extensive revisions to the manuscript as a result of the valuable comments and suggestions from both reviewers. In addition to a review of the revised manuscript for syntax and grammar, the manuscript has been checked to ensure it meets all journal requirements.

To facilitate review, all suggested changes, as well as additional revisions to the manuscript, are in red font. We hope our manuscript is now acceptable for publication in PLOS ONE. Our responses are as follows:

Additional journal requirements

Response: We have checked the manuscript to ensure that it meets PLOS ONE's requirements for style and file naming.

2. Please describe in your methods section how capacity to provide consent was determined for the participants in this study. Please also state whether your ethics committee or IRB approved this consent procedure. If you did not assess capacity to consent please briefly outline why this was not necessary in this case.

Response: The caregivers provided confirmation that the persons with dementia understood the purpose of the study and were willing to participate. We have added this information to the methods (Pages 6-7, lines 134-139). The IRB and the heads of the outpatient clinics approved this study (Page 6, lines 126-128).

Response: A minimal anonymized dataset has been uploaded as Supplemental data (S1 File. Minimal anonymized dataset).

Responses to the comments from Reviewer 1

Comment 1: This is a well-timed article addressing serious and pressing concerns in relation to ACP and people with dementia for end of life care. An issue which is increasingly concerning given the increasing prevalence of dementia in Asia and globally. The intervention described serves a purpose even if this raises awareness among FCG's. The obvious detriment is the ability of a person with dementia being able to retain information due to memory loss. The article discusses that studies have shown that interventions regarding ACP can increase awareness and decision making between PWD, FCG's and clinicians.

However, the article omits to mention where there is also a lack of congruency in such discussions (see Jones et alones, K., Birchley, G., Huxtable, R., Clare, L., Walter, T., & Dixon, J. (2016). End of life care: A scoping review of experiences of Advance Care Planning for people with dementia. Dementia: the international journal of social research and practice. DOI: 10.1177/1471301216676121.

Response 1: Thank you for your suggestion. We have added the following to Page 4 (lines 71-73) of the introduction:

Moreover, systematic literature reviews revealed that proxies were often unable to accurately predict patients’ treatment preferences, especially for patients with stroke or dementia [15, 16]. 

Comment 2. I was also curious about the issue of one off vs ongoing consent that might be expected in a study involving PWD. For example, securing consent and the right to withdraw right up to the stage when data is amalgamated and anonymised and it is no longer possible to withdraw data.

Response 2: Thank you for your concern. This study was conducted with dyads of persons with dementia and the family caregiver. We have added additional information describing ongoing monitoring of the ability of the persons with dementia to participate (Page 7, lines 150-157):

All participants provided written informed consent prior to the pretest; persons with dementia provided consent for collection of medical data from their charts. We also invited caregivers to confirm that the person with dementia understood the purpose of this study and were willing to participate. If any person with dementia showed distress, appeared distracted, or provided an answer that was not relevant to the question a reminder about the question or a short break was offered. If the response of the participant did not improve, the interview ceased. If either the person with dementia or their caregiver withdrew or terminated participating in the research, all data collection ceased for the dyads.

Comment 3: There is also the issue that PWD with dementia having received a diagnosis of early stage dementia as identified by the use of the MMSE and CDR and other information, may in fact find it hard to imagine their future selves as and having to make decisions around oral hydration for example.

Response 3: Thank you for your concerns. Indeed, as you said, most people with mild dementia or their caregivers do not understand the end-stage symptoms of dementia and find it hard to imagine their future selves as and having to make decisions regarding end-of-life care. It is for this reason that the first part of the intervention is to provide the participants with information about the symptoms of end-stage dementia and the type of care involved, such as a tube feeding option for eating difficulties. Below is the description of the information provided during the ACP Intervention (Page 8, lines 166-169): 

The nurse explained the contents of the manual, which included descriptions about the symptoms of end-stage dementia and the common EoL life-sustaining medical treatments, such as CPR, machine ventilation, tube feeding, intravenous infusion, and antibiotics. The manual also provided details about the benefits and risks of the treatments, as well as the process of establishing ACP and the regulations involved.

Comment 4: On page 25. there is a typo under the heading After effects of the intervention on decisional conflicts (line 2) 'The result was is'. If the authors could clarify.

Response 4: Thank you for your correction. We have deleted “is”. 

Comment 5. I am in agreement with the recommendations that as part of the ACP interventions clinicians and nursing professionals should be trained to facilitate this tool. Overall, I would attend to these queries which are few as this is a good article.

Response 5: Thank you for your comment that this is a good article and we hope our intervention will become a valuable reference for clinical professionals. We appreciate all your thoughtful comments and suggestions, which we believe have strengthened our findings.

Response to the comments from Reviewer 2 (Dr. van der Steen)

Comment 1: There is not much evidence about advance care planning effects in case of early dementia, and therefore a non-controlled pre-post test design does add substantially to the current body of evidence. Most parts of the study are well-designed, individual patient summed scores are being provided, and the article is generally well written.

Response 1: Thank you for these comments. Indeed, it is the limitation of this study that there is no control group and only one group for pre-posttest design. We have listed this as a limitation and hope to improve this problem in future studies.

Comment 2: A main concern is inclusion of persons with no dementia diagnosis but just mild cognitive impairment. This is not only a methodological problem (which may be overcome by sensitivity analyses excluding the persons with MCI). Because the intervention explains disease prognosis to persons who may not have and may not develop the disease, this is also an ethical concern. There was an IRB approval of the local hospital for this intervention study. In some other countries though, monitoring of such study might be as stringent as for drug trials because of the vulnerability of the population and the sensitive nature of advance care planning discussions. Further, did the informed consent form include permission to provide the participants’ data open access?

Response 2: Thank you for your comments and suggestions. Whether persons with mild cognitive impairment eventually develop dementia has indeed been discussed in the literature, and this was also a concern of one of the other reviewers. We have added this as a limitation to our findings (Page 25, lines 456-464):

Next, we only recruited persons with mild cognitive impairment and mild dementia for this study. Excluding persons with moderate dementia prevents the generalization of our findings to all persons with dementia in Taiwan. In addition, our hypothetical situations used to test decision conflicts are not necessarily representative of real-life situations and therefore, the findings should be interpreted with caution. Persons with mild cognitive impairment do not necessarily develop dementia, and the perspectives of persons with mild cognitive impairment regarding decisions about ACP and EoL care might differ from persons with dementia. A more heterogenous sample of persons with dementia should be considered for future studies. 

Regarding your concern about the ethics of discussing the disease prognosis with those who might not develop dementia, only hypothetical statements or questions were used for the survey data. This was explained to the participants. 

The informed consent form of this study indicated the results would be anonymized and would only be published in academic literature. All personal data was protected by coding and there was no display of the participants’ names or personal data. We appreciate your mention of open access. Unfortunately, a request for permission to provide open access to the data was not included in the consent form. If the IRB determines this does not prevent us from providing an anonymized dataset, we the data will be uploaded as a supplementary file. 

Comment 3: Please avoid abbreviating persons by their disease. See for example Alzheimer Australia for language guidelines.

Response 3: Thank you for this comment. We have changed the abbreviation from PWDs to persons with dementia throughout the manuscript. 

Comment 4: The attitude outcome is very interesting. Unfortunately, it is not clear from the cited reference what the items look like. Ref 29 shows 8 negative and 5 positive items, different from the numbers in the paper.

Response 4: We thank the reviewer for this question. The original scale of attitudes towards the Taiwan’s Hospice-Palliative Care Act included 8 pro- and 6 con- items, which were provided by Dr. Hu (the corresponding author of Ref 25). We added one additional pro-item and two additions con-items. We have provided more detail about the scale as it was used in this study in the methods (Page 11, lines 226-231):

The original scale included a total of 14 items describing the pros (positive attitudes, 8 items) and cons (negative attitudes, 6 items) of Taiwan’s Hospice-Palliative Care Act. Our research team added one pro-item related to increasing communication about ACP between patient, family and the medical team and two con-items related to ACP procedures and payments. The 17 items were scored on a 5-point Likert scale from (1) strongly disagree to (5) strongly agree. 

Comment 5: The study uses the validated decisional conflict scale as a relevant outcome. The knowledge outcomes are not much interesting, as the items of the scales resemble the information provided in the intervention about invasive medical procedures, knowledge that most people lack (e.g. explain CPR and test knowledge about CPR). What is interesting and new though, is that the knowledge test was also completed by persons with dementia. Did you reassure persons with dementia if they felt they failed the test, did you have any protocol for this?

Response 5: We thank the reviewer for this question. We have provided additional information about the testing process, which allowed the person with dementia additional time and flexibility to help them feel more comfortable with the testing. First, in the section describing the design and study population (Page 7, lines 152-157) we added the following:

We also invited caregivers to confirm that the person with dementia understood the purpose of this study and were willing to participate. If any person with dementia showed distress, appeared distracted, or provided an answer that was not relevant to the question a reminder about the question or a short break was offered. If the response of the participant did not improve, the interview ceased. If either the person with dementia or their caregiver withdrew or terminated participating in the research, all data collection ceased for the dyads.

For knowledge of end-stage dementia treatment (Page 10, lines 206-208) we added the following:

Both persons with dementia and FCGs were reminded that if they were not sure if the statement was true or false to please answer “unknown”.

For the decisional conflict scale (Pages 11-12, lines 244-253), we added the following:

Each item of the scale was a question about EoL care decision options to either accept or forgo: CPR, machine ventilation, and tube feeding. The question format is easier for participants with limited reading skills [38]. Items were scored on a 5-point Likert scale: from (1) strongly agree to (5) strongly disagree. If the participant indicated that they had no actual decision-making ability, the research assistant asked them to indicate what their choice would be if they needed to make an EoL care decision right now. For example, a question on the subscale of effective decisions, states “I feel I have made an informed choice about EoL care for myself (person with dementia)/or for my relative (FCG)”. If the participant was unwilling to make a hypothetical choice, the question was marked as not applicable, and the calculated score was weighted based on the total answers for the subscale.

Comment 6: Introduction: “for individuals with moderate to severe dementia, life expectancy ranges from 1.5 to 2.5 years [4-6].” Reference 6, however, depicts a median survival of about 5 years for MMSE 0-18.

Response 6: Thank you for your correction. We have deleted reference 6.

Comment 7: Introduction: “the decline is irreversible and becomes more severe” is not accurate. Actually, there is evidence of the reverse, the decline, which is of course a rate becoming less steep towards the end as people can linger on with advanced dementia for a long time (e.g. Gill et al., NEJM) although caution as findings may be related to sensitivity or ceiling effects of some dementia severity measures in the more severe spectrum.

Response 7: Thank you for this suggestion. We are in complete agreement that not all persons with dementia experience severe impairment. However, a decline in cognitive functions is an indicator that the disease has worsened. We have tempered our statement about the severity of the disease (Page 3, lines 58-63), which we agree is not as steep as indicated by older studies. This sentence now reads: 

As the disease advances, the trajectory for persons with dementia is characterized by a gradual decline in memory, cognition, activities of daily living (ADLs), and decision-making abilities. With the exception of dementia due to an underlying illness, such as chronic hypertensive encephalopathy, hepatic dysfunction, or vitamin deficiency, which account for less than 10% of all dementias, the decline is irreversible and cognitive abilities decrease with the progression of the disease [7, 8]. 

Comment 8: Introduction: please correct “a randomized control study”

Response 8: Thank you for your noticing our error. This has been corrected as “A randomised controlled study”, (Page 4).

Comment 9: Introduction: family members cannot be held responsible for advance care planning, in most countries physicians decisions are ultimately decisive even after shared decision making as they cannot be requested to provide medically futile care. 

Response 9: Thank you for this comment. We have revised the introduction to better explain the need for shared decision-making (Pages 5-6, lines 106-113):

ACP directives emphasize the value of an individual’s personal autonomy and preferences, thus, in many western countries, family members are not responsible for decisions about ACP. However, family harmony and interdependency are highly valued in Taiwanese culture, particularly when facing end-of-life care decisions [25], which makes it less likely an individual will initiate an advance directive. The increase in the prevalence of dementia in the general population has drawn attention to the need for shared decision-making between family caregivers and persons with dementia as a component of dementia care [26, 27]. 

Comment 10: Methods: need to provide a cut off and a proper methodological reference for the Kuder-Richardson statistic.

Response 10: Thank you for your suggestion. We did not set a cut-off point, but we have added a reference related to the KR item difficulty and item discrimination (Page 10, lines 209-213):

In this study, the KR value for persons with dementia and FCGs were 0.83 and 0.81 respectively, indicating an adequate and acceptable homogeneous test [35]. The ranges of item difficulty and item discrimination for persons with dementia (0.27–0.55 and 0.45–1.00, respectively) and FCGs (0.27–0.73 and 0.27–0.91, respectively) were within acceptable ranges [36]. 

Comment 11: Methods: how did respondents complete the subscales Uncertainty and Effective decision if no decision was taken or if there were more decisions? The items of the other subscales are more general, but the items of these 2 subscales refer to a particular decision. Did you provide any instruction to the respondents, or else, how did you manage while cleaning data?

Response 11: Thank you for these questions. We added the following explanation (Page 12, lines 247-253):

If the participant indicated that they had no actual decision-making ability, the research assistant asked them to indicate what their choice would be if they needed to make an EoL care decision right now. For example, a question on the subscale of effective decisions, states “I feel I have made an informed choice about EoL care for myself (person with dementia)/or for my relative (FCG)”. If the participant was unwilling to make a hypothetical choice, the question was marked as not applicable, and the calculated score was weighted based on the total answers for the subscale. 

Comment 12: Results: a total of 86 persons in 40 dyads of whom there were 2 family caring for 2 patients, would result in a total of 78 persons. Please explain how you arrived at 86, does it include persons who refused or did not provide meaningful data? How many did?

Response 12: Thank you for your reminder. We have corrected the number (Page 13, lines 271-273):

Therefore, a total of 78 persons (40 dyads) completed the intervention; two caregivers cared for two persons with dementia, thus they were paired with the two persons with dementia repeatedly for a total of 40 persons with dementia and 38 FCGs. 

Comment 13: Table 1: please avoid too many decimal places that indicate fractions of persons with these modest numbers as it suggests a precision that is not meaningful. For example, 19 of 40 persons is shown as 47.5%. If it were 18 persons, it was 45%. Rounding off to 48% is best as 47.4% or 47.6% is impossible, even 46% is. I know you can still see it in the literature, and unfortunately, the PWD is widespread too, but let’s try to do better and professionalize our area of research.

Response 13: Thank you for your suggestion. We have rounded the percentages in the Tables. Please see Table 1 and Pages 13-14, lines 282-285.

Comment 14: Results: “positive score increased slightly (33.00 → 34.98)” Please do not overinterpret. The 2 decimals with no SD mentioned in the text are misleading for the magnitude and uncertainty around this “slight effect.”

Response 14: Thank you for this comment. We have deleted “however the positive score increased slightly (33.00 → 34.98)”. We revised the sentence (Page 14, lines 293-294) as follows:

No significant change was observed in either the positive (33.00 → 34.98) or negative (23.03 → 22.93) attitude towards ACP. 

Comment 15: Results: “was significantly lower for PWDs (49.88)” A more accurate description of the result would be to add the comparator with posttest lower than pretest.

Response 15: Thank you for this suggestion. The sentence (Pages 14-15, lines 294-296) has been revised:

The mean total score for posttest decisional conflicts in EoL care was significantly lower than pretest for persons with dementia (49.88 → 35.11, t = 4.76, p < .001, effect size = −0.8). 

Comment 16: Results: there was a reduction in the uncertainty of decision making. Which decisions? Please add type and number of decisions made to the Results section.

Response 16: The authors thank you for your suggestion. We have clarified that the five subscales for decisional conflict also decreased significantly (Page 15, lines 296-300):

All five subscale scores were also significantly lower: informed (55.00 → 40.83, t= 2.88, p < .01, effect size = -0.06), values clarity (58.96 → 41.46, t= 3.76, p < .01, effect size = -0.08), support (48.54 → 36.11, t= 3.86, p < .001, effect size = -0.06), uncertainty (45.63 → 30.42, t= 4.37, p < .001, effect size = -0.067), and effective decision (43.44 → 29.22, t= 4.05, p < .001, effect size = -0.06).

On Page 17 (lines 314-315), we added the following:

The decreases for uncertainty and effective decision were 37.50 → 29.58 and 37.50 → 29.06, respectively.

Comment 17: Methods/discussion: What does family-centered mean? Do you mean to say it is not patient-centered even though you choose a population of patients with a mean of about 20 so probably very capable of their preferences being at the center. If this is a cultural issue, please explain as it is relevant for western cultures if the intervention translates to a more individualized context.

Response 17: The authors thank you for your suggestion. Indeed, Chinese cultures are more family oriented and care is family-centered rather than person-centered. We have revised the first paragraph of the Discussion to put this in context (Page 19, lines 321-329):

Although ACP emphasizes the patient’s autonomy and is designed to be person-centered, family members are usually expected to be proxies in Chinese countries. If the FCGs do not understand the content of an ACP for the patient, they are often unable to adhere the patient’s preferences. The Taiwanese Patient Right to Autonomy Act, Article 9 stipulates that “the declarant, at least one relative of first or second degree of affinity, and the health care agent shall participate in the advance care planning” [39]. The purpose of this procedure is to increase communication and promote a harmonious relationship between the patient, family and medical professionals. Therefore, the family-centered ACP intervention model may be more suitable for countries with a Chinese culture that is family-centered, such as Taiwan.

Comment 18: Discussion: “A second explanation might be due to the ACP attitudinal scale used in this study, which was a revision of the attitudinal scale of the Hospice-Palliative Care Act [29].” What is the explanation? Please explain.

Response 18: The authors thank you for your question. After re-reading this section of the discussion we realized additional clarification was required for this explanation. We have revised the text (Page 21, lines 375-382):

A second explanation might be due to the ACP attitudinal scale used in this study, which was developed previously to reflect attitudes towards the Hospice-Palliative Care Act [25] legislated in Taiwan in 2000. The Patient Right to Autonomy Act was not officially launched until January 2019, therefore participants may not have been familiar enough with the new regulations involving ACP to express an opinion. If the participants did not express any positive or negative opinions regarding ACP during the intervention, there was no opportunity for the facilitator to address these attitudes and values and provide any clarification. 

Comment 19: Discussion: “Our study is unique in that persons with mild dementia who retained decision-making abilities were purposively sampled to participate.” Two observations: not all had dementia. Further, purposive sampling is an expression used mostly for qualitative research to indicate purposeful sampling of a non-representative sample to elicit rich information. I cannot see how this term applies to this particular study other than that there was an inclusion criterion for this dementia-specific dyadic intervention with limitations in and of itself because persons with moderate dementia were excluded and persons with MCI were included.

Response 19: The authors thank you for this comment. You are correct in that purposive sampling is primarily used in qualitative research. We recruited persons with mild dementia because the study was designed determine if the ACP intervention would be effective for dyads of persons with mild dementia and family caregivers. We have deleted “purposive sampling” and revised the methods as follows (Page 6, lines 131-134):

Dyads of persons with mild dementia and their FCGs were recruited from outpatient clinics of regional teaching hospitals, dementia care centers, or community care centers located in Taipei City and New Taipei City, Taiwan. 

We have revised the wording in the discussion (Page 23, lines 413-426):

This study is unique in that is the first assessment of the effectiveness of an intervention to increase knowledge and facilitate information sharing about ACP between persons with mild dementia and FCGs. We recruited persons with mild cognitive impairment or mild dementia who retained decision-making abilities in order to optimize the initial evaluation of the intervention. During the ACP intervention process, FCGs could help persons with dementia understand that ACP allows one to share their values and wishes prior to losing their decision-making abilities. In addition, because the FCGs were familiar with the characteristics and habits of the persons with dementia, the intervention also helped FCGs identify the key concerns. The companionship and support provided by the FCGs may have helped the persons with dementia feel more at ease, which is a crucial factor for enabling independent decision-making [50-52]. Future studies will evaluate whether this intervention is effective for persons with greater levels of cognitive impairment based on scores on the CDR or MMSE, which could provide information on a cutoff point where the intervention is not helpful. 

We have also added the exclusion of persons with moderate dementia as a limitation of our study (Page 25, lines 456-458):

Next, we only recruited persons with mild cognitive impairment and mild dementia for this study. Excluding persons with moderate dementia prevents the generalization of our findings to all persons with dementia in Taiwan.

---

## [Decision Letter · Decision Letter 1]

1 Oct 2020

Advance Care Planning Information Intervention for Persons with Mild Dementia and Their Family Caregivers: Impact on End-of-Life Care Decision Conflicts

PONE-D-20-21167R1

Dear Dr. Huang,

We’re pleased to inform you that your manuscript has been judged scientifically suitable for publication and will be formally accepted for publication once it meets all outstanding technical requirements.

Kind regards,

Stephen D. Ginsberg, Ph.D.

Section Editor

PLOS ONE

**Comments to the Author**

1. If the authors have adequately addressed your comments raised in a previous round of review and you feel that this manuscript is now acceptable for publication, you may indicate that here to bypass the “Comments to the Author” section, enter your conflict of interest statement in the “Confidential to Editor” section, and submit your "Accept" recommendation.

Reviewer #1: All comments have been addressed

2. Is the manuscript technically sound, and do the data support the conclusions?

Reviewer #1: Yes

3. Has the statistical analysis been performed appropriately and rigorously? 

Reviewer #1: I Don't Know

4. Have the authors made all data underlying the findings in their manuscript fully available?

Reviewer #1: Yes

5. Is the manuscript presented in an intelligible fashion and written in standard English?

Reviewer #1: Yes

6. Review Comments to the Author

Reviewer #1: The authors have responded to comments and have amended the article per reviewer 1 as requested and against the following criteria:

1. The study presents the results of original research.

2. Results reported have not been published elsewhere.

4. Conclusions are presented in an appropriate fashion and are supported by the data.

5. The article is presented in an intelligible fashion and is written in standard English.

6. The research meets all applicable standards for the ethics of experimentation and research integrity.

7. The article adheres to appropriate reporting guidelines and community standards for data availability.

Reviewer 2 is responding to statistical analysis.

7. PLOS authors have the option to publish the peer review history of their article (what does this mean?). If published, this will include your full peer review and any attached files.

Reviewer #1: No

---

## [Editor Report · Acceptance letter]

5 Oct 2020

PONE-D-20-21167R1 

Advance care planning information intervention for persons with mild dementia and their family caregivers: Impact on end-of-life care decision conflicts 

Dear Dr. Huang:

I'm pleased to inform you that your manuscript has been deemed suitable for publication in PLOS ONE. Congratulations! Your manuscript is now with our production department. 

Kind regards, 

on behalf of

Dr. Stephen D. Ginsberg 

Section Editor

PLOS ONE